# The Elusive Quest for Valuation of Coastal and Marine Ecosystem Services

**J. Walter Milon** [1,*] **and Sergio Alvarez** [2]

[1]   Department of Economics, University of Central Florida, Orlando, FL 32816, USA
[2]   Rosen College of Hospitality Management, University of Central Florida, Orlando, FL 32816, USA
*   Correspondence: jmilon@ucf.edu

**Abstract:** Coastal and marine ecosystem (CME) services provide benefits to people through direct goods and services that may be harvested or enjoyed in situ and indirect services that regulate and support biological and geophysical processes now and in the future. In the past two decades, there has been an increase in the number of studies and journal articles designed to measure the economic value of the world's CME services, although there is significantly less published research than for terrestrial ecosystems. This article provides a review of the literature on valuation of CME services along with a discussion of the theoretical and practical challenges that must be overcome to utilize valuation results in CME policy and planning at local, regional, and global scales. The review reveals that significant gaps exist in research and understanding of the broad range of CME services and their economic values. It also raises questions about the validity of aggregating ecosystem services as independent components to determine the value of a biome when there is little understanding of the relationships and feedbacks between ecosystems and the services they produce. Finally, the review indicates that economic valuation of CME services has had a negligible impact on the policy process in four main regions around the world. An alternative direction for CME services research would focus on valuing the world's CME services in a wealth accounting framework.

**Keywords:** coastal; marine; ecosystem services; economic valuation; wealth accounting; public policy

---

## 1. Introduction

Coastal regions around the world have been important centers for population growth and economic development, and this trend is projected to continue throughout the 21st century [1]. Although coastal areas cover only 4% of the Earth's total land area and 11% of the ocean area, they contain more than a third of the Earth's human population and are more than twice as densely populated as inland areas [2,3]. The land–sea interface and the coastal and marine ecosystems (CMEs) within this interface provide a rich array of goods and services ranging from fish and shellfish harvesting in seagrass beds, mangroves, oyster reefs, and coastal bays and estuaries to recreation and tourism on beaches, shores, and coral reefs. Past development, however, has led to significant loss and degradation of these CMEs [4,5]. Costanza et al. [6] estimated that the global supply of CMEs, with the exception of seagrass beds, had exhibited significant losses in both area and total economic values between 1997 and 2011. However, recent research indicates that seagrass beds have also experienced a marked global decline in terms of area covered [7]. Future demographic trends and sea-level rise will intensify the pressures for change in these ecosystems [1].

A number of national and international policies have been implemented to protect and restore CMEs and the ecosystem services they provide [8–12]. Following the Millennium Ecosystem Assessment [2] framework, these ecosystem services provide benefits to people through direct goods and services that may be harvested or enjoyed in situ and indirect services that regulate and support

biological and geophysical processes such as nutrient cycling, water purification, and reproductive habitats now and in the future [13]. A number of different classification systems have been developed in recent years to define the linkages between ecosystem services and benefits to people including the National Ecosystem Services Classification System [14], the Common International Classification of Ecosystem Services [15], and the United Nations System of Environmental-Economic Accounting [16]. In addition, the International Organization for Standardization has recently issued ISO 14,008, which specifies a methodological framework for the monetary valuation of environmental and related impacts on human health, the built environment, and ecosystems [17].

A critical component in identifying the contribution of ecosystem services from CMEs is quantification of these contributions as economic values or similar metrics. Economic valuation is increasingly recognized as a necessary component of CME evaluation and policy analysis because it provides a commensurate metric to compare different ecosystem services and tradeoffs between these services and other economic activities that may impact CMEs [8]. In the past two decades, there has been an increase in the number of studies and journal articles designed to measure the economic value of CMEs, although there are significantly fewer published coastal and marine articles than for terrestrial ecosystems [18,19]. Access to the results of these studies is now available through online databases such as the National Ocean Economics Program/Middlebury Institute of International Studies at Monterey, Economics of Ecosystems and Biodiversity, Environmental Valuation Reference Inventory, and the Gulf of Mexico Ecosystem Services Valuation Database. In addition, ecosystem services models such as the Integrated Valuation of Ecosystem Services and Tradeoffs (InVEST) have modules that can be applied for CMEs.

In this article, we provide a review of the literature on valuation of CME services along with a discussion of the theoretical and practical challenges that must be overcome to utilize valuation results in CME policy and planning at large scales. We begin in the next section with an overview of the different coastal and marine ecosystems and the provisioning, cultural, regulating, and supporting services provided by these ecosystems. We also relate these services to specific types of direct, indirect, and nonuse values for each CME. Section 3 reviews the economic research to value CME services in terms of the focus on specific types of CMEs and services. This review indicates that past research has focused heavily on coral reefs and mangroves and the provisioning services they provide. Other CMEs such as seagrass beds and oyster reefs have not received as much attention, and the values associated with regulating and supporting services for CMEs are not well understood. Section 4 considers the role of valuation in the CME policy and planning process in four regions across the world and the impediments to successful integration of CME services valuation. The final section considers future directions for CME services valuation and the difficult challenges associated with a complete accounting for the Earth's CME services.

## 2. The Scope of Coastal and Marine Ecosystem Services and Economic Valuation

Coastal and marine ecosystems and the processes they support produce flows of services that are highly valuable to society [3,14]. As long as they are not degraded or depleted, these ecosystems will continue to produce services that offer a variety of benefits to people. These benefits can be measured in monetary units using total economic values (TEVs) that recognize the direct, indirect, and nonuse contributions of these ecosystem services to humanity e.g., [20]. However, this monetary valuation may be difficult and may not capture the full range of ecological, socio-cultural, and non-anthropocentric values that can be ascribed to natural capital such as CMEs [21,22]. Nevertheless, one advantage of monetary valuation is that it can potentially be used within a framework that consistently evaluates both conventional economic activity and changes in the stock of natural capital within a unified wealth accounting system e.g., [16,23,24]. Similarly, monetary valuation of ecosystem services can be used to examine the impact of specific policies or management interventions affecting these ecosystems through benefit–cost analysis [25].

*2.1. Ecosystem Services in the Economic Valuation Framework*

The conceptual work emanating from the Millennium Ecosystem Assessment [2] was among the first efforts to define a comprehensive classification system for the types of services provided by ecosystems. These ecosystem services can be grouped into four categories: provisioning, cultural, regulating, and supporting. Provisioning services provide food and raw materials that can be directly consumed or utilized by humans. Cultural services are nonmaterial benefits that are enjoyed through recreation and aesthetic experiences, spiritual, or artistic appreciation. Regulating services have a less direct (albeit no less important) impact on humans through processes such as water treatment or purification, carbon storage, hydrologic regulation, flood protection, wave attenuation, and erosion prevention. Supporting services are necessary for the production of other ecosystem services and include natural habitats, biological diversity, and climate stability. These categories are not mutually exclusive, and the continuing supply of ecosystem services depends on healthy interaction and maintenance of all stocks of natural capital.

The TEV is the most common valuation framework to capture the range of benefits from ecosystem services, and it provides an overview of the benefits that humans receive from ecosystems and their services as well as the motivations that people may have for wanting to conserve and preserve ecosystems. In other words, TEV is an anthropocentric concept that considers economic value strictly as physical or perceived benefits to humans, a point which has fueled disagreements between economists and conservationists on the issue of intrinsic value [26]. TEV is the sum of benefits that derive from direct use value (UV), indirect use value (IUV), and nonuse value (NUV). These can be further refined into more specific subcategories e.g., [20], but these distinctions are most relevant in the consideration of future supply of ecosystem services. In the context of CMEs, UVs come from the harvesting of fish, wildlife, and raw materials in different CME habitats as well as activities that involve non-consumptive recreational uses such as snorkeling, scuba diving, and nature observation. Regulating services such as storm protection and erosion control provide IUVs. Supporting services such as biodiversity and ecological connectivity that allow species to move across habitats at different stages of their life cycle are not directly or indirectly consumed but provide NUV through their role in maintaining ecosystem functions and the future availability of ecosystem services.

A more complete classification of economic values for ecosystem services from different types of CMEs is presented in Table 1. For each category of ecosystem services, UV and IUV benefits are noted in regular font and NUVs are in italics. The geographic scale at which these benefits are typically received is denoted along the bottom of the table. The most commonly recognized benefits are provisioning and cultural services that are directly used, most often, at the local or regional scale. As discussed in the following section, these are the ecosystem services from CMEs that have been most widely studied for economic valuation. Regulating services such as the storm protection benefits of mangroves, beaches, and dunes also provide UV and IUV and have received some attention in the valuation literature. Other types of regulating services of CMEs such as providing nursery and protective habitat for a range of fish and wildlife species are less understood and more difficult to value. Typically, these services function at a broader regional scale, so identifying the 'extra-local' linkages across ecosystems is necessary for services valuation e.g., [27].

**Table 1.** Coastal ecosystems and types of economic values for ecosystem services. Direct and indirect use values in regular type, nonuse values in italics.

| Coastal Ecosystem | Ecosystem Services | | |
|---|---|---|---|
| | Provisioning and Cultural | Regulating | Supporting |
| Coral Reefs | Recreation and tourism, fish and shellfish harvesting, raw materials, *education* and *aesthetics* | Storm protection, nutrient cycling | *Biological diversity, ecological connectivity, habitat for fish and shellfish, nursery and protective habitat* |
| Seagrass beds and salt marshes | Fish and shellfish harvesting, raw materials, wildlife harvesting, *education* and *aesthetics* | Storm protection, erosion control, water purification, oxygen cycling, nutrient cycling, carbon storage and sequestration | *Biological diversity, ecological connectivity, nursery and protective habitat for fish, shellfish and wildlife* |
| Mangroves | Fish and shellfish harvesting, raw materials, *education* and *aesthetics* | Storm protection, nutrient cycling and erosion control, water purification, oxygen cycling, carbon storage and sequestration | *Biological diversity, ecological connectivity, nursery and protective habitat for fish, shellfish and wildlife* |
| Oyster reefs | Shellfish harvesting, raw materials, *education* and *aesthetics* | Storm protection, erosion control, water purification, nutrient cycling, carbon storage and sequestration | *Biological diversity, ecological connectivity, nursery and protective habitat for fish and shellfish* |
| Beach, dune and shore | Recreation and tourism, *education* and *aesthetics* | Storm protection, erosion control | *Biological diversity, ecological connectivity, nursery and protective habitat for shellfish and wildlife* |
| Bays and estuaries | Recreation and tourism, fish and shellfish harvesting, raw materials, wildlife harvesting, *education* and *aesthetics* | Storm protection, erosion control, water purification, oxygen cycling, carbon storage and sequestration, nutrient cycling | *Biological diversity, ecological connectivity, nursery and protective habitat for fish, shellfish and wildlife* (see [4] p. 187) |
| | *Local* | **Scale** | *Global* |

At the broadest, or global scale, supporting services from CMEs produce NUV for processes such as biodiversity and carbon sequestration. These are ongoing processes at the regional and global scale that are necessary for the long-term survival of CMEs. These processes, however, are difficult to characterize, and there is more uncertainty about their functioning. For example, biodiversity improves the capacity of ecosystems to adapt to climate change, but the changes in species composition and density may be unknown [28]. Spatial mapping is one tool that can be used to identify linkages between ecosystem services across different geographic scales, e.g., [29]. A major limitation for spatial mapping of CME services is that, unlike terrestrial ecosystems that can be mapped with remote sensing or satellite imagery, there is a scarcity of spatial data to effectively address the dynamic nature of coastal and marine environments across both spatial and temporal dimensions [19,30]. In other words, mapping what is under the surface of the ocean is much more challenging than mapping systems that

can be readily seen by the naked eye. However, novel methods such as multibeam echosounder [31] and the IKONOS satellite sensor [32] are improving our capacity to map CMEs.

*2.2. The Economic Valuation Process*

From a valuation perspective, the basic steps to measure the TEV from CME services appear to be relatively straightforward [4,33]. First, identify the specific ecosystem service of interest and the underlying ecosystem functions and processes that produce that service. In the case of fish and shellfish provisioning from saltwater marsh and seagrass beds, for example, this would involve the biological processes (such as recruitment and growth) and the ecological requirements (such as suitable habitat and water quality) that produce these harvestable products for recreational and/or commercial users, e.g., [34]. Second, determine how existing ecological conditions contribute to production of the ecosystem services and, in the case of potential management actions or other events that would change the underlying conditions, how flows of ecosystem services could change in the future. In the context of salt marshes or seagrasses, for example, the valuation exercise would consider how changes in the area of these habitats and/or changes in salinity conditions within these habitats would impact fish and shellfish production. Third, utilize economic valuation methods to measure the TEV from the existing, and/or potential, level(s) of ecosystem services.

In general, the objective of economic valuation is to provide estimates of the net present value of ecosystem services, and a number of prior papers review the range of economic valuation methods for CMEs, e.g., [35–38]. Values for ecosystem services can be derived directly using information from prices of related market goods and services, such as by using current prices of fish products to estimate future losses in fisheries due to ocean acidification, e.g., [39]. However, most ecosystem services benefit humans in ways that are not captured in existing markets, so nonmarket valuation methods are necessary for these cases. Broadly, nonmarket valuation methods can be divided into revealed preference and stated preference methods. With revealed preference methods such as the travel cost and hedonic pricing methods, information on human behavior in existing markets, such as travel or housing, are used to estimate the value of related ecosystem services, such as beach recreation (by observing preferences for people to travel to beaches) or clean waterways (by observing price premiums for homes adjacent to clean water). In contrast, stated preference methods such as contingent valuation or discrete choice experiments rely on surveys designed to elicit people's preferences for specific ecosystem services. In recent years, studies have also used estimates of the 'social cost of carbon' to build estimates of the net present value of services related to carbon storage. The major difference between this approach and the market and nonmarket methods is that the social cost of a carbon approach does not rely on studying people's preferences, but rather it relies on simulations of the global economy under different climate change scenarios.

The specific context in which economic valuation may be utilized can significantly add to the complexity of the process. In marine planning, for example, Borger et al. [8] describe a ten-step process that begins with recognition of the initial planning authority, stakeholder participation, evaluation of existing and future conditions, and monitoring and adaptive management. In the context of the U.S. National Ocean Plan and various coastal and marine planning initiatives in the U.K., the opportunities to integrate valuation in the planning process can be very different [8]. Similarly, the European Marine Strategy Framework Directive has resulted in a mixed record of success for applications of economic valuation because of the varying requirements of the planning process [12,40].

## 3. Towards a Comprehensive Valuation of Coastal Ecosystem Services

*3.1. Valuing the Biosphere's Services*

Costanza et al.'s [41] seminal paper began a series of discussions that continue to this day on the value of the world's ecosystem services and natural capital, the feasibility of estimating these values at a global scale, and the adequacy of existing methods and data for constructing these estimates.

The original analysis relied on the use of a value or benefits transfer meta-analysis of the literature existing at the time on 17 ecosystem services across 16 biomes, which assumed a constant value per unit area for each biome. The most striking finding was an estimate of the entire biosphere's ecosystem services ranging between USD$ 16–54 trillion ($10^{12}$) per year, with a mean of USD$ 33 trillion per year. CMEs represented 68% of the total value, or nearly USD$ 22.4 trillion. In comparison, the study reports the value of global gross national product at the time to be USD$ 18 trillion per year. Thus, the main message was that the world's ecosystems, and CMEs alone, were more valuable than the direct market goods and services produced by the entire global economy.

At its core, this approach relies on the construction of databases of existing valuation studies that include estimated monetary value, the original units of measure, the method used to develop the estimate, the type of value estimated, the year of estimation, the original currency of the estimate, the characteristics of the ecosystem or biome, and the ecosystem service valued [42]. These databases can then be used to calculate measures such as the total or average estimated value of services produced by specific biomes or estimates of the value of ecosystem services produced per unit area for specific biomes. The databases can also be used to construct regression equations that can serve as benefit transfer functions to estimate the value of particular ecosystem services or biomes at different scales.

A series of important studies followed in the wake of Costanza et al. [41] with expansions and updates of the concept. For example, Braat et al. [43] developed the Cost of Policy Inaction (COPI) Valuation Database to quantify the global social and economic costs from the past, present, and future loss of biodiversity. They estimated that by 2050 humankind is expected to lose land-based ecosystem services worth an estimated €14 trillion, with losses in marine and coastal ecosystem services of a comparable magnitude. In contrast to the annual loss figures reported by Costanza et al. [41], the losses reported by Braat et al. [43] are cumulative losses of ecosystem services between 2008 and 2050. Similarly, de Groot et al. [42] report on progress made through the compilation and publication of The Economics of Ecosystems and Biodiversity's (TEEB) Ecosystem Service Valuation Database (ESVD), which built on the COPI database and was developed to be usable by different stakeholders to estimate values for 10 main biomes in a common spatial, temporal, and currency unit (international dollars per hectare per year). In a similar vein, Plantier-Santos et al. [44] developed the Gulf of Mexico Ecosystem Service Valuation Database (GecoServ) as an inventory of ecosystem services valuation studies applicable to the Gulf of Mexico.

Costanza et al. [6] expanded on the global approach by accounting for changes in the (per hectare per year) value of ecosystem services produced by specific biomes through time and changes in the global area covered by different biomes. The tabulation quantified the change in the value of ecosystem services between 1997 and 2011 as an annual loss ranging between USD$ 4.3–20.2 trillion. In addition, the reported values per unit area of CMEs increased between 1997 and 2011 based on new studies in the literature. However, the overall value of CMEs declined to 59.6% of the total value of global ecosystems after adjusting for both changes in unit value and areal declines, most notably in coral reefs and wetlands.

*3.2. Limitations of the Global Valuation Approach*

While these global-scale ecosystem valuation studies have raised public awareness about the magnitude of the contribution of ecosystems and their services to human societies, the approach has many limitations. Several researchers have pointed out that one major problem is the available data to conduct benefit transfers at a global scale are much too limited as they come from a relatively small number of primary studies over a limited geographic area. Pendleton et al. [35] were among the first to conduct a literature review on the value of CMEs. Based on over 100 studies in the U.S., they reported the valuation estimates were predominantly concentrated on four CMEs: coastal beaches, marine fishing, coastal lands, and wetlands. Also, the geographic coverage was concentrated in a few states such as Florida, California, and the Carolinas, but had scant coverage in other regions such as the Pacific Northwest, the Great Lakes, and New England.

Barbier [3] and Barbier et al. [4] provided a more global perspective on the CMEs valuation literature. These reviews reported that only coral reefs, salt marshes, and mangroves had been studied in any detail, and even these studies focused on a small subset of ecosystem services mostly related to provisioning. Barbier et al. [4] also noted a second major problem in that most studies focused on ecosystem services as independent products with little consideration of ecological interactions between services or across different ecosystems. Therefore, there is no valid basis to aggregate CME service components to determine the value of a biome if there is no understanding of the effects of relationships and feedbacks between ecosystem services (ES) components.

The Exxon Valdez oil spill in Alaska in 1989 and the Deepwater Horizon spill in the Gulf of Mexico in 2010 triggered a number of economic valuation studies focused on the impacted coastal and marine ecosystems. Most of these studies, however, relied on benefit transfers from other studies and focused mostly on provisioning services such as recreation [45]. If new research was conducted, it also focused on recreational activities, e.g., [46]. When a broad array of CME services was addressed, the analysis typically used a replacement cost approach, habitat equivalency analysis, so the results were not applicable for other areas beyond the damage zone [45]. Another focus area has been the use of valuation for coastal management decisions, particularly related to establishment of marine protected areas such as the National Marine Sanctuaries, or broadly on the value of specific ecosystems such as coral reefs or beaches. However, it has been noted that, "The variability in context, methodology, and approach in these studies do not allow for aggregation and make comparability of these resulting value estimates difficult" [45].

These prior reviews of the CME valuation literature raise concerns about the limited number of primary studies for a broad array of CME services and the lack of knowledge about how CME services may be interrelated. These two major problems can be further illustrated with the global valuation data from de Groot et al. [42] and McVittie and Hussain [47]. Table 2 presents mean and median valuation estimates for coastal and marine biomes that were used in Costanza et al. [6] and represent the existing literature at the time. Also presented in Table 2 are the percentages of the mean estimates attributed to specific categories of ecosystem services and the number of studies (in parentheses below) reported for each category.

**Table 2.** Ecosystem service components as a percent of total mean value (in 2007 Int.$/ha/year) for coastal and marine biomes; number of value estimates in parentheses below each percent. Adapted from: [42,47].

| Biome | Total Mean (Median) Value $/ha/Year | Provisioning | Cultural | Regulating | Supporting |
|---|---|---|---|---|---|
| Coral reefs | $352,249 ($197,900) | 15.8% (47) | 30.9% (53) | 48.9% (24) | 4.6% (13) |
| Coastal wetlands | $193,845 ($12,163) | 1.6% (80) | 1.1% (20) | 88.5% (30) | 8.9% (12) |
| Other Coastal systems | $28,917 ($26,760) | 8.3% (23) | 1.0% (10) | 89.4% (5) | 1.3% (7) |
| Open Ocean | $491 ($135) | 20.8% (7) | 65.0% (5) | 13.2% (1) | 1.0% (3) |

The valuation estimates for coral reefs in Table 2, the most studied (137 value estimates) of the CME groups, indicate it is the highest valued ($352,249 $/ha/year) with a broad range of value attributed to different ecosystem services. Regulating and supporting services comprise a majority share of the mean value estimates, although there is no way to know how an increase in provisioning services (through heavier use or harvest) would either increase or decrease the values attributed to other ecosystem services. The open ocean, on the other hand, is the least studied (16 value estimates) biome and has a mean valuation ($491 $/ha/year) that is a fraction of the value attributed to coral reefs. The vast array of ecosystem services from the open oceans (e.g., fisheries, deep water habitats, climate regulation,

biodiversity, and carbon sequestration) has simply not been studied from a valuation perspective. Also, independent valuation by biome provides no recognition of the interconnectivity between open oceans and other biomes, such as coral reefs, that depend on ocean transport processes that distribute nutrients, larvae, and adult species.

For coastal wetlands and other coastal systems, the data in Table 2 indicates that the vast majority of benefits from these CMEs were derived from regulating and supporting services. There are a variety of services from these CMEs such as nutrient cycling, coastal protection (through attenuation of wave action), erosion control (through stabilization and retention of soil/organic matter and sediments), water purification (through uptake and retention of nutrients), and carbon sequestration (by fostering the accumulation of organic matter). Yet, these value estimates were based on a small number of studies, many of which were conducted more than two decades ago [42].

Criticisms of the global aggregation approach to ecosystem valuation are not new. For instance, Bockstael et al.'s [48] criticism provides an overview of the challenges inherent in a comprehensive valuation of the services provided by the biosphere's ecosystems. Economic valuation of ecosystem services entails a choice made by individuals where two alternative states of the world are weighed against each other, and each state is associated with a different level of ecosystem services being provided (however, valuing all of the world's ecosystems entails a choice where one alternative includes the biosphere as we know it, and the other entails either the absence of the biosphere's ecosystems or a state of the world that we cannot realistically define. Even if such a choice could be defined in a way that individuals could understand, their willingness (and ability) to pay to avoid such a loss of ecosystems cannot exceed their income, so the value of the world's ecosystem services cannot—by definition—exceed the combined gross income of all the people of the planet.). Pendleton et al. [49] summarized these shortcomings in a review of Costanza et al. [6] with the observation that, "despite the efforts devoted to the evaluation of the world's ecosystem services over the last three decades, these data generally are insufficient to do much more than raise awareness. In fact, the limitations of the results produced by Costanza et al. [6] illustrate the enduring lack of accurate and comprehensive, global data for ecosystem services—especially for marine and coastal areas" [49].

A major challenge in the quest for comprehensive valuation of CMEs (and ecosystem services in general) is the integration of ecological production functions into the estimation of economic valuation functions [4,33,50]. Successful valuation of changes in ecosystem services requires establishing a clear connection between an ecosystem and the ecosystem services it provides or between the ecological production function and the economic valuation function [4]. To make this connection, the changes in ecosystem structure, function, and processes that give rise to the changes in ecosystem services must be characterized. This characterization is a necessary step before a clear linkage between the changes in ecosystem structure, function, and processes, and the changes in the quantities and qualities of ecosystem service flows to people can be made. Without a clear connection, economic valuation methods are unlikely to yield meaningful and unbiased estimates.

Given that complete economic valuation of ecosystem services relies on adequate understanding of ecosystem structure, function, and processes, and their links to the flow of ecosystem services that benefit humans, valuation of benefits arising from highly complex characteristics of the natural world can be an especially challenging endeavor. A case in point is the economic valuation of biological diversity, a complex, multilevel concept that includes genetic, species, functional, molecular, and phylogenetic diversity. But, the meaning of biodiversity in political discourse and economic valuation has not been precise. Because there is no single 'right' indicator of biodiversity, economic valuation studies designed to estimate the value of biodiversity have used different proxies such as numbers of species, single species or groups of species, genetic diversity, functional diversity, habitats, or even more abstract measures such as 'low', 'medium', or 'high' biodiversity [51]. Thus, compiling value estimates from all available studies on the value of biodiversity in order to extract an average value for this complex concept is troublesome. This problem of complex processes and characteristics as valuation objects is not unique to biodiversity but a feature of supporting ecosystem services

in general. Similar issues have been found in the valuation of cultural ecosystem services, where cultural ecosystem services are defined in a variety of ways [52], and researchers have recognized that the current typologies of cultural ecosystem services are insufficient to account for services such as ingenuity, life teaching, and perspective [53].

The interrelationships between multiple services produced by the same ecosystem must also be carefully considered before attempting to add up individual ecosystem service values to obtain an estimate of the ecosystem's total value. These relationships and feedbacks between ecosystem services, ecosystem components, and the ecosystems as a whole are what was previously referred to as the ecosystem production function [4,33,50]. While an ecosystem may simultaneously produce multiple services, some uses of these services may diminish the capacity of the ecosystem to produce other services. For example, use of coastal ecosystems by recreational vessels provides important cultural ecosystem services to people, but it may cause decreased water quality, introduction of non-native species, and physical disturbances [54]. Ecosystem deterioration resulting from these impacts of recreational use can be expected to alter the structure and function of the ecosystem and, thus, reduce the provision of other ecosystem services.

In the case of seagrass beds, for instance, recreational boating may lead to introduction of pathogens, pollutants, or invasive species that can disrupt the life cycle of native organisms. Similarly, high boating traffic may result in physical disturbance of the seafloor, which will result in deterioration of the sea grass beds and thus result in lower levels of regulating services such as water purification, carbon sequestration, and erosion control. Thus, there may be conflict between different types of ecosystem services where humans' use of one service may hinder production of other services. In some cases, use of cultural and provisioning services may cause deterioration or reduction in regulating and supporting services. Conversely, enhancement of a particular ecosystem service may result in complementary enhancements to other services produced by the same ecosystem. For example, restoration of oyster reefs to mitigate the effects of eutrophication also results in the creation of habitat for finfish.

Most importantly, linear changes in the areal extent of ecosystems may result in nonlinear changes to the provision of ecosystem services. For instance, linear increases in the size of mangrove forests result in quadratic increases in wave attenuation services [55]. Thus, simply adding up the values of ecosystem service components to obtain a total value for the ecosystem is likely to result in misleading estimates. Similarly, linear multiplication of areal extent of an ecosystem by an average value per area to obtain an estimate of the total value of an ecosystem relies on a strong assumption of linearity and will likely lead to biased estimates.

## 3.3. Current Status of Coastal and Marine Ecosystem (CME) Valuation

More recent reviews of the literature on CME valuation [18,38] show that research has continued moving in a similar direction, and thus the problems highlighted previously are still a primary concern. Torres and Hanley's [18] global review found a total of 196 studies published on CME services valuation between 2000 and 2015. The most studied CME during this period was beaches, with 40 papers published. Conversely, the least studied CME was the deep ocean, with only two papers published during this period. Most studies lacked a comprehensive approach to capture both use and nonuse values, although some recent research has integrated valuation for a broader array of CME services using stated choice valuation methods [18].

The recent literature also highlights the need for improved CME services valuation information in the study of climate change impacts on human societies and economies [56]. The value of CME services can be expected to be heavily impacted by sea level rise and extreme weather events, which will reshape the world's coastlines and the areal extent of many CMEs. In addition, warmer ocean temperatures will have significant impacts on primary productivity and the physiology of organisms inhabiting the world's oceans [34]. However, with some exceptions, e.g., [39], most of the existing approaches to estimate the future impacts of climate change rely on integrated assessment models

(IAMs) that are used to estimate a social cost of carbon (SSC) based on the expected changes in social welfare associated with different levels of carbon emissions [31,57,58]. Most of these IAMs currently ignore the nonmarket values produced by CMEs and their services [56].

Better harmonization between IAMs and nonmarket values requires a four-step procedure. First, global climate models are downscaled to a case study area where likely changes in climate can be related to physical changes in CMEs. Second, these changes are linked to specific changes in ecosystem services that are currently provided by CMEs in the study location. Third, qualitative changes in provision of affected ecosystem services are specified in quantitative terms. Last, these quantitative changes in ecosystem service provision can be valued using existing methods [38,59].

## 4. Valuation in Coastal and Marine Policy

Valuation of ecosystem services provided by CMEs can inform a myriad of policy decisions, including assessment of wetland mitigation strategies, design of erosion prevention and coastal conservation programs, natural resource damage assessment, fisheries management, and design of biodiversity conservation strategies, among others. In addition, valuation work can also provide important insights related to the design of financing mechanisms, including payment for environmental services schemes, as well as the implementation of environmental legislation.

### 4.1. Potential Uses of Valuation in the Policy Process

The use of valuation in policy and planning decisions can be defined as either informative or decisive [60]. Informative uses of ecosystem services valuation are intended to contribute to discussions and draw attention to specific resources and/or ecosystem services. For example, global studies of the total economic value of all ecosystems, e.g., [6,41] were intended principally to promote awareness and interest in the importance of ecosystem services. While these studies elevated the visibility of ecosystem services in both public and academic discourse, they relied on prior valuation studies and benefit transfer methods that lacked precision and did not address the types of problems encountered in coastal and marine ecosystem management [49,61].

Decisive uses of valuation studies in policy, on the other hand, seek to evaluate the tradeoff between different levels and/or types of ecosystem services. For example, fishery managers may set different total allowable catch limits for a coastal species that determine the amount of provisioning services from current harvests and the supporting services from future stock. Or, coastal managers may want to evaluate strategies to maintain or expand existing wetlands to sustain the storm protection benefits of wetlands and prevent future land loss from erosion and hurricanes, e.g., [62]. These tradeoffs are regional or local, and the relative benefits and costs could be compared with valuation information. A similar policy problem is the tradeoff between ecosystem services from natural capital and development. The conversion of an estuarine salt marsh to commercial or residential property will reduce the fishery production, wildlife habitat, and storm protection benefits from the salt marsh. Estimates of the value of these ecosystem services would provide a direct economic comparison of the associated benefits and costs of alternative uses for decision making.

In the following, we review some examples of the use of ecosystem services valuation in coastal and marine policy across four geographic and political regions: the European Union, the United States, Australia, and the Caribbean. We focus on the legislation or directives that may require or encourage the use of ecosystem services valuation (ESV) in planning decisions, the available literature on CME service values, and the reported success to date in the actual integration of ESVs into coastal and marine policy.

### 4.2. The European Union

One example of regional multinational policy setting for the use of CME services valuation is the European Marine Strategy Framework Directive established in 2008 and the Maritime Spatial Planning Directive in 2014. These directives require European Union member states to take measures to achieve

or maintain "good environmental status" in the marine environment by 2020 [63]. Ecosystem-based management has been recognized as an analytical approach that establishes a broad framework to assess coastal resource and human interactions. These two directives, and other more resource-specific coastal and marine national and European legislation [63], provide for the evaluation of indicators, targets, and economic analyses to develop programs to achieve this objective and to consider transboundary features and impacts [12,40,64]. Formal guidance has been developed to utilize an ecosystem services based approach, and some results have been achieved in compiling studies and inventories for different CME service values, e.g., [65].

Despite the broad scope of the directives, there has been limited success in actual implementation of ecosystem services valuation in the planning and evaluation process. Drakou et al. [64] reviewed 11 European case studies and found that valuation studies did inform decision makers. In only a few cases, however, were the valuation results applied or used to influence decision making. The authors recommend greater involvement of 'end users' in planning, development of more integrated frameworks for coastal and marine social–ecological systems, and better informing the general public about the role of ecosystem services. Similarly, several authors [12,40,66] reported that the most common problems in the use of valuation studies involved a general lack of knowledge and expertise with ecosystem service and management concepts, gaps in understanding how changes in ecosystem functions and services impact human welfare, limited valuation data for a broad array of CME services, a lack of involvement by diverse end user and stakeholder groups in the policy planning process, and a lack of time and resources to effectively coordinate ecological and social research and planning across diverse spatial and temporal scales.

### 4.3. The United States

US federal government involvement in coastal and marine planning typically requires economic analyses consistent with the Water Resources Council's Principles and Guidelines established in 1983. These guidelines did not recognize ecosystem services as a component of evaluation, but the 2013 Principles, Requirements and Guidelines for Federal Investments in Water Resources [67] required an explicit recognition of ecosystem services in program management and planning [68]. In addition, the National Ocean Policy Implementation Plan [69] established a comprehensive regional approach to CME management in the US. This policy initiative recognized the linkage between CMEs and economic well-being, and it focused on expanding public awareness and data accessibility for information about the economic values associated with the broad spectrum of use and nonuse values from CME services for use in coastal and marine planning [8]. Recent revisions to the National Ocean Policy, however, may jeopardize prior directives to focus on a broad perspective for economic values from CME services in coastal and marine planning [70].

Despite these new policies, there are few reported applications of ecosystem valuation in coastal and marine resource policy and planning decision-making in the U.S. The National Centers for Coastal Ocean Science has undertaken research to estimate ecosystem values related to several estuarine reserves using a variety of valuation methods [71–73]. In addition, the issues related to state and local coastal and marine planning, where the majority of coastal planning choices occur, have been identified, and research needs have been defined, e.g., [74,75]. But, there is no published research to document whether or how ecosystem services valuation information has been used in coastal and marine policy or planning in the U.S.

### 4.4. Australia

In Australia, federal guidelines encourage the use of benefit–cost analyses for major regulatory programs that would impact coastal and marine resources, but actual valuation of ecosystem services is not required [66]. Despite the lack of an official policy context for valuation, considerable literature on CME services valuation for Australia has emerged [76,77]. Based on a survey of decision-makers in local, regional, state, and national government agencies as well as representatives of marine

industries, Marre et al. [77] reported that ESVs were most often used in management decisions involving commercial fisheries, recreational activities and tourism, and coastal development/conservation.

Specific examples of ESV having a significant impact on decision-making were for marine park zoning in the Great Barrier Reef, e.g., [78] and protection strategies for seagrass meadows [79]. In general, however, CME service values were used mostly to communicate and support evaluation for ecosystem services. Most decision-makers believed that valuation information had a weak influence on coastal and marine policy [77]. A recent study by Sandhu et al. [80] in South Australia utilized a scenario planning process to engage a diverse group of stakeholders and decision-makers to anticipate potential future changes in economic growth, land cover, and ecosystem services from alternative coastal development pathways. While the process revealed important differences across the scenarios, there was no discussion on whether the results were integrated into regional coastal decision making.

### 4.5. The Caribbean

The region of the world with perhaps the greatest volume of studies on coastal and marine ecosystem valuation is the Caribbean. Schuhmann and Mahon [81] identified over 100 studies with valuation results for CME services with the majority focusing on coral reefs and marine protected areas. The quantity of research is partly attributable to efforts by the World Bank to support valuation studies in the 1990s and the Caribbean Large Marine Ecosystem project initiative by the United Nations Development Programme beginning in 2009. The latter explicitly adopted an ecosystem-based management approach to promote regional development and sustain coastal and marine ecosystem processes.

Waite et al. [82] conducted a literature review and personal interviews with coastal resource managers, government officials, and others throughout the Caribbean to determine the extent to which CME services valuation studies had been used in decision making. They reported that, in general, valuation studies had little impact on the policy process. They did identify, however, a small group of 'success stories' where the information directly influenced decision-making across several countries. The common factors that were found across these successful cases were mainly procedural and methodological and included: a clear intended use (policy question) for valuation, strong stakeholder engagement, access to decision-makers, and potential revenue enhancement for the government. Despite the limited influence from prior CME services valuation studies, the authors suggest that awareness of, and demand for, the information provided from CME services valuation is growing and could play a greater role in policy and planning in the Caribbean in the future.

## 5. Summary and Discussion

### 5.1. Summary

The development and dissemination of the ecosystem services framework in the early 2000s created a new research agenda for valuation research. The new paradigm broadened the scope of valuation concerns for coastal and marine resources from previous research that had focused almost exclusively on provisioning services such as harvesting and recreation. This new perspective gave greater recognition to a broader range of ecosystem services beyond traditional provisioning, and it put greater attention on nonuse values that had received little attention in the literature. This was especially challenging in light of the relative scarcity of data and limited understanding of spatial interconnectivity in coastal and marine ecosystems compared to terrestrial systems, e.g., [19].

This systematic review of the literature on CME services valuation indicates that studies of provisioning services still dominate valuation research. While the TEEB valuation inventory summarized in de Groot et al. [42] was notable in its attempt to expand awareness of ES components, the compilation reveals the gaping holes that exist in research and understanding of the broad range of values for CME services. It also raises serious questions about the validity of simply aggregating

ES components to determine the value of a biome when there is little understanding of the effects of relationships and feedbacks between ES components on total values.

These gaps largely reflect the fact that most CME studies took a piecemeal approach and were 'one shot' efforts for a specific problem or setting. Since recreation- and market-based activities are easier to observe and measure, use values are the most obvious to estimate. Regulating and supporting services are more obscure and typically require nonuse valuation methods. Most studies lacked a comprehensive approach to capture both use and nonuse values, although some recent research has integrated valuation for a broader array of CME services [18]. In addition, these studies rarely take into consideration the impacts that regulation and management may have on the value of interconnected provisioning and supporting ecosystem services [83].

Part of the reason for this piecemeal approach to CME services valuation is the general lack of any broader research initiatives to move beyond one-time valuation studies. As noted by several authors, e.g., [8,67,84], CMEs research and valuation studies have not been a funding priority, and policymakers oftentimes do not understand the role of ES research. Recent projects have also emerged to develop simpler, less expensive simulation tools such as InVEST, e.g., [30,85] in place of valuation research for specific coastal and marine planning problems. A major shortcoming, however, is that these policy-directed tools almost exclusively utilize benefit transfer methods and do not address issues of measurement error and validity. The valuation information from these policy tools are, on a positive note, second-best shadow values or, more likely, no more than informative indicators to recognize the role of CME services.

Despite the increased focus in the scientific community on the importance of ES, coastal and marine resource policy and planning decisions typically do not require ES valuation. Also, resource managers and decision-makers often lack a basic understanding about ES values. A review of studies from four major geographic regions across the globe indicates that CME service values are most often used for informational purposes and rarely used to evaluate tradeoffs for policy decisions. The studies cite a variety of reasons for this situation, but the most common concerns are a lack of understanding of ES for both policymakers and the public and the limited availability of valuation information for non-provisioning services in specific settings. This decision-making context for CME services policy and planning is a classic 'wicked problem' with complexity, interdependence, and conflicting social interests [22].

Some have argued that focusing on 'benefit-relevant indicators' for CME services in the planning process can provide useful information when full economic valuation is not practical or possible for the specific planning problem [86,87]. Others have argued that conventional TEV cannot account for the full range of sociocultural and ecological values from CME services. They propose a broader array of participatory and stakeholder involvement approaches for a more complete evaluation of ecosystem values in the planning process, e.g., [84,88–90].

While these alternative approaches may help in specific management settings in the short run, they do little to further a comprehensive understanding of values for a full range of CME services. Typically, these participatory approaches focus on local problems, and the participants may not reflect broader public interests, e.g., [66]. This is especially a concern in the European Union setting where the supply of ecosystem services is characterized by high degrees of spatial transboundary interdependence, e.g., [63]. Economic analysis may be difficult to apply for all CME services, but it can provide a consistent framework to develop valuation information and to evaluate that information across national boundaries.

*5.2. Alternative Directions*

The current approach to ES valuation for coastal and marine policy and planning is fraught with shortcomings. An alternative is to expand on the traditional national economic accounts that provide objective, regular, and standardized information that public and private interests rely on for planning and decision making [91]. A broader system of 'wealth accounting' would include

natural capital accounts (NCAs) and ecosystem goods and service accounts (EGSAs) that could provide governments and businesses with information on the current status of a nation's (or the entire biosphere's) natural capital and ecosystem services [23,92,93]. This wealth accounting framework is not new, e.g., [94,95], but the ecosystem services paradigm has renewed international attention on the merits of a comprehensive approach to environmental valuation [24]. This alternative also does not negate the need for, and importance of, smaller scale valuation studies that link ecological production functions to specific economic benefits for coastal and marine policy decision making, e.g., [81].

For coastal and marine policy and planning, adopting the wealth accounting framework is an ambitious undertaking. But, the conceptual and practical frameworks have been defined. For example, a wealth accounting framework would include changes in fishery stocks as part of national income in much the same way as changes in farmed animal stocks are now included. The reason why fish stocks are excluded is because they are considered natural assets beyond the production boundary of the current system, whereas livestock is a produced asset controlled by human activities. Guidelines for implementing this approach for fisheries have existed for many years, e.g., [96], and additional guidance to expand the framework for a broad range of natural assets and ecosystem services is also available, e.g., [16].

The primary advantage of the wealth accounting framework is that it provides consistent, internationally recognized standards for valuation [17,91]. A major challenge for applying this framework to coastal and marine resources is that many of the ecosystem services are nonmarket outputs and often provide nonuse values. Provisioning and cultural services are fully amenable to wealth accounting, but regulating and supporting services must be tracked through to the final consumption of households distinct from market goods. This requires a separate accounting of nonmarket services consumption that can highlight the relative contribution of different ecosystem services to human welfare [24,97].

The information that can be available for coastal and marine policy and planning from a wealth accounting approach is important to inform the public about the relative importance of CME services and to evaluate tradeoffs between different types of CME services. For example, Barbier [92] demonstrates how households' willingness to pay for restoration of estuarine and coastal habitats is directly related to the expected future storm protection benefits from these restored habitats. Similarly, if extraction activities to harvest the provisioning services available in the short run from a CME threaten sustainability of a basket of other supporting and regulating services in the long run, then the total value of the CME will be degraded in the future. A coral reef, for example, that provides short-term benefits from the provision of recreation and harvesting but is over-utilized or degraded due to human visitation, may experience reduced levels of fishery habitat and biodiversity services in the future. A comprehensive valuation framework is needed that recognizes these tradeoffs and the capacity of the CME to generate these ecosystem services in the future. Successful implementation of a wealth accounting framework will require a commitment by national governments across the world to fully inform the public about the important services provided by natural capital such as CMEs.

The ecosystem services paradigm has changed the terms of debate about coastal and marine valuation for policy and planning. There have been clear inroads in public recognition of the role of ecosystem services, and there are some examples around the world where CME services valuation information has had an impact on policy and decision making. The question looking forward is whether this new ecosystem services perspective can change the direction of research on CME services valuation and the information available to decision-makers and the public.

**Author Contributions:** Conceptualization, J.W.M.; methodology, J.W.M.; software, J.W.M., S.A.; validation, J.W.M., S.A.; formal analysis, J.W.M., S.A.; investigation, J.W.M.; resources, J.W.M., S.A.; data curation, J.W.M., S.A.; writing—original draft preparation, J.W.M., S.A.; writing—review and editing, J.W.M., S.A.; visualization, J.W.M., S.A.; supervision, J.W.M.; project administration, J.W.M.

**Funding:** This research received no external funding.

**Conflicts of Interest:** The authors declare no conflict of interest.

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
