# Peer review of "The Elusive Quest for Valuation of Coastal and Marine Ecosystem Services"

_water, doi:10.3390/w11071518_

Round 1

Reviewer 1 Report

This manuscript delivers high-quality review for such a problematic and complex issue on obtaining the adequate valuation of coastal and marine ecosystem services. Admittedly, the manuscript is quite dense and rich of information, but with a well-organized structure; all the acronyms and terms are explained at the right place, and it builds on gradually so that even a non-expert reader can follow and get the key points.

There is a potential for further discussion about the problems behind the lack of inclusion of actual valuation studies in the decision-making process. An underlying cause that could be read out of the examples is the lack of accountability within the decision-makers and policymakers on one side with the ever-decreasing trust of stakeholders and the general public in the institutions managing the ecosystem resources and services. However, such discussion might go in a different direction than this manuscript is referring to.

Along with a critical presentation of current flaws in the domain of CMEs valuation, this review also delivers potentially promising new directions and, while keeping unbiased attitude, opens the door to new areas or research.

Few minor points that may be addressed prior to publication are listed below:

Authors keep using the citation as a subject in the sentence, sometimes at the right beginning of it (e.g. Page 1, line 37 – “[6] estimated…”. Same goes for Page 3, line 96; Page 5, line 175, and multiple occasions after). This is rather inconvenient and distracts a reader who has to check the reference each time, because if the reference is subject, one is to expect that Authors/Study case is rather essential to have in mind instead of the [x].

URLs for references 9 and 10 are not valid

Using both trillion and billion expressions for apparently same number (10^12) on page 5 (line 181-182) is unnecessarily confusing

Page 5, line 199, does this €14 trillion goes for a yearly amount of generated value (as compared to values in the first paragraph of this section), or as a total value of the lost system?

Page 9-10, MSFD and MSPD are annotated as Directives, hence it is not clear on which other Directives are Authors referring to in line 403. Short clarification or example of Directive is recommended (e.g. Bathing Water Quality, or Habitats, The Common Fisheries Policy etc.).

Author Response

This manuscript delivers high-quality review for such a problematic and complex issue on obtaining the adequate valuation of coastal and marine ecosystem services. Admittedly, the manuscript is quite dense and rich of information, but with a well-organized structure; all the acronyms and terms are explained at the right place, and it builds on gradually so that even a non-expert reader can follow and get the key points.

There is a potential for further discussion about the problems behind the lack of inclusion of actual valuation studies in the decision-making process. An underlying cause that could be read out of the examples is the lack of accountability within the decision-makers and policymakers on one side with the ever-decreasing trust of stakeholders and the general public in the institutions managing the ecosystem resources and services. However, such discussion might go in a different direction than this manuscript is referring to.  Along with a critical presentation of current flaws in the domain of CMEs valuation, this review also delivers potentially promising new directions and, while keeping unbiased attitude, opens the door to new areas or research.

Thank you for this thoughtful review.

Few minor points that may be addressed prior to publication are listed below:

 Authors keep using the citation as a subject in the sentence, sometimes at the right beginning of it (e.g. Page 1, line 37 – “[6] estimated…”. Same goes for Page 3, line 96; Page 5, line 175, and multiple occasions after). This is rather inconvenient and distracts a reader who has to check the reference each time, because if the reference is subject, one is to expect that Authors/Study case is rather essential to have in mind instead of the [x].

Thank you for making this suggestion.  We have revised the manuscript to mention author names when possible.

URLs for references 9 and 10 are not valid

We have checked the URLs and they still link to the public-facing EPA websites containing the referenced reports.

Using both trillion and billion expressions for apparently same number (10^12) on page 5 (line 181-182) is unnecessarily confusing

Corrected, all values expressed as trillion.

Page 5, line 199, does this €14 trillion goes for a yearly amount of generated value (as compared to values in the first paragraph of this section), or as a total value of the lost system?

In contrast to the annual loss figures reported by Costanza et al. [36], the losses reported by Braat et al. [38] are cumulative losses of ecosystem services between 2008 and 2050.

This sentence has been added to clarify.

Page 9-10, MSFD and MSPD are annotated as Directives, hence it is not clear on which other Directives are Authors referring to in line 403. Short clarification or example of Directive is recommended (e.g. Bathing Water Quality, or Habitats, The Common Fisheries Policy etc.).

Given the number of directives and laws that could be referred to by this sentence, we have added an additional reference where they are described in more detail.

Reviewer 2 Report

Review of: The Elusive Quest for Valuation of Coastal and Marine Ecosystem Services

for: Water

Overview

The authors review the literature on the valuation of coastal and marine ecosystem (CME) services, providing historical context for the evolution of that vein of research. The economic concept of (total economic) value is covered, and a nice table explaining the classes (provisioning, cultural, regulating, supporting) of ecosystem services provided by various CMEs is provided. A large portion of the paper is devoted to the discussion of global-scale values of CMEs, but more local applications are discussed as well. The authors discuss some of the challenges of attempting to assess values on a global scale and of estimation of values on a more local scale, and also deliver the unfortunate news that valuation does not seem to be an important input into the decision-making process for policy-makers.

I enjoyed reading this paper and think it would provide some useful information and an important perspective for readers of the journal. I have a few things that I’ll suggest to the authors.

In my opinion, there’s too much focus on global-scale valuation and not enough on local/project-scale valuation, which is probably more useful.

Section 3, which takes up several pages, covers valuation of CME services on a global scale, heavily citing the work of Costanza and colleagues. As both the authors and Constanza et al. (2014) state, this kind of global assessment of the values of CME services is probably most useful for “raising awareness and interest” on the importance of CMEs and the large role they play in human welfare. I don’t think many readers will really be interested in these global-scale assessments. I could be wrong, of course. But I think if the authors are going to take up so much space with this discussion, they should explain why it’s so important, how is this type of analysis useful, etc. I think one reason I don’t care much for them is that “value” in economics has meaning only by comparing two different states of the world. At a local, project-level, it’s easier to imagine the world with and without the project in place, estimate what the physical changes will be (e.g. changes in ecosystem services), and then estimate the value of those physical changes. At a global scale, what is the alternative state of the world? No ecosystem services whatsoever anywhere on earth? That scenario doesn’t have much meaning for me.

On the other hand, the authors discuss more meaningful (to me) project-level valuation studies relatively little. E.g. The authors could discuss other valuation methods—they mainly focus on benefits transfer through meta-analyses—like contingent valuation, travel cost methods, hedonic pricing, etc. This would help readers of the journal understand that there are a bunch of different valuation techniques that they can consider when teaming up with an appropriately trained economist or two. With this discussion, studies using these techniques can be cited for further reference for the reader.

The authors correctly point out that one of the difficulties with valuation of CME services is that the changes in CME services (say, from a project or policy) have to be assessed, then these changes have to be linked to stuff people care about, and then the changes in stuff people care about can be valued. Well, this connection is rather meaningless on a global scale just because it would be so unwieldly, but on a project-level scale the connection can be more clear.

Relatedly, how does the macro-level stuff in section 5.2 help? What can it be used for? I’m not very big on macro, but I guess like GDP it might be some measure of how well-off nations are?

All this is to say that if the authors spend so much time discussing stuff on large scales, they could help the reader by also explaining why this type of assessment is useful. But regardless, it seems they don’t discuss small scale valuation, which certainly has been the application of most CME valuation studies, enough.

Can the authors make a more clear recommendation about the relationship between valuation and policy?

This is something I always struggle with myself, and I think the authors say it well: from an economic point of view we want to compare benefits and costs of a policy or project to see if it’s a good idea, but policy-makers don’t think the same way generally, and so any estimates economists come up with for benefits or costs (as related to CME service changes) don’t really end up playing a role in policy decisions.

So are valuation economists wasting their time? It gets a bit depressing, I think. I really do like how the authors highlight instances where valuation did play more of a role in policy, and what it was about the considered studies that helped them play a more effective role in policy. The authors devote a few sentences to this, but I think it’s important enough to add a few more details, like specifics of what made the studies accessible to policy-makers (if the authors want to go this route, that is). That way the reader gets some clear recommendations about how his or her work can be more relevant.

The last two paragraphs before section 5.2 have a bit of circular reasoning I wish the authors could help me work through. On the one hand, using TEV “cannot account for the full range of sociocultural and ecological values from CME services” (so, it’s implied, we can’t rely solely on economic concepts of what makes a policy good or bad and we need to consider other criteria), but on the other hand, using these other non-economic criteria “[does] little to further a comprehensive understanding of values for a full range of CME services.” I guess the authors are just telling it like it is, but I feel like I want them to give me a recommendation (I being an economist) about how to deal with the reality of decision-making while still thinking like an economist and comparing costs and benefits (which requires TEV estimation).

Other comments:

·         Table 1 is great.

·         p.8 line 344, It’s pretty obvious that the S in CMES stands for services, but I don’t think CMES was ever defined.

·         p.11 line 488, I believe “relatively” should be “relative”.

Author Response

The authors review the literature on the valuation of coastal and marine ecosystem (CME) services, providing historical context for the evolution of that vein of research. The economic concept of (total economic) value is covered, and a nice table explaining the classes (provisioning, cultural, regulating, supporting) of ecosystem services provided by various CMEs is provided. A large portion of the paper is devoted to the discussion of global-scale values of CMEs, but more local applications are discussed as well. The authors discuss some of the challenges of attempting to assess values on a global scale and of estimation of values on a more local scale, and also deliver the unfortunate news that valuation does not seem to be an important input into the decision-making process for policy-makers.

I enjoyed reading this paper and think it would provide some useful information and an important perspective for readers of the journal. I have a few things that I’ll suggest to the authors.

Thank you for your thoughtful review.

In my opinion, there’s too much focus on global-scale valuation and not enough on local/project-scale valuation, which is probably more useful.

The paper is a review of global or large scale valuation efforts.  Our intent is to provide an overview of recent developments in this area, and review its promise and pitfalls.  We have revised the title and the manuscript as a whole to ensure that this scope is clear.

While we agree that local/project-scale valuation is more reliable and probably more useful than global-scale valuation, a review of local/project scale valuation is beyond the scope of the paper.

Section 3, which takes up several pages, covers valuation of CME services on a global scale, heavily citing the work of Costanza and colleagues. As both the authors and Constanza et al. (2014) state, this kind of global assessment of the values of CME services is probably most useful for “raising awareness and interest” on the importance of CMEs and the large role they play in human welfare. I don’t think many readers will really be interested in these global-scale assessments. I could be wrong, of course. But I think if the authors are going to take up so much space with this discussion, they should explain why it’s so important, how is this type of analysis useful, etc. I think one reason I don’t care much for them is that “value” in economics has meaning only by comparing two different states of the world. At a local, project-level, it’s easier to imagine the world with and without the project in place, estimate what the physical changes will be (e.g. changes in ecosystem services), and then estimate the value of those physical changes. At a global scale, what is the alternative state of the world? No ecosystem services whatsoever anywhere on earth? That scenario doesn’t have much meaning for me.

We have added the following paragraph to this section to address this comment:

“Criticisms of the global aggregation approach to ecosystem valuation are not new.  For instance, Bockstael et al.’s [43] criticism provides an overview of the challenges inherent in a comprehensive valuation of the services provided by the biosphere’s ecosystems.  Economic valuation of ecosystem services entails a choice made by individuals where two alternative states of the world are weighed against each other, and each state is associated with a different level of ecosystem services being provided.A 

AHowever, valuing all of the world’s ecosystems entails a choice where one alternative includes the biosphere as we know it, and the other entails either the absence of the biosphere’s ecosystems or a state of the world that we cannot realistically define.  Even if such a choice could be defined in a way that individuals could understand, their willingness (and ability) to pay to avoid such a loss of ecosystems cannot exceed their income, so the value of the world’s ecosystem services cannot—by definition—exceed the combined gross income of all the people of the planet.”

On the other hand, the authors discuss more meaningful (to me) project-level valuation studies relatively little. E.g. The authors could discuss other valuation methods—they mainly focus on benefits transfer through meta-analyses—like contingent valuation, travel cost methods, hedonic pricing, etc. This would help readers of the journal understand that there are a bunch of different valuation techniques that they can consider when teaming up with an appropriately trained economist or two. With this discussion, studies using these techniques can be cited for further reference for the reader.

While we agree that local/project-scale valuation is more reliable and probably more useful than global-scale valuation, a review of local/project scale valuation is beyond the scope of the paper.  These concerns have been addressed explicitly with footnote A.

The authors correctly point out that one of the difficulties with valuation of CME services is that the changes in CME services (say, from a project or policy) have to be assessed, then these changes have to be linked to stuff people care about, and then the changes in stuff people care about can be valued. Well, this connection is rather meaningless on a global scale just because it would be so unwieldly, but on a project-level scale the connection can be more clear.

While we agree that local/project-scale valuation is more reliable and probably more useful than global-scale valuation, a review of local/project scale valuation is beyond the scope of the paper.  These concerns have been addressed explicitly with footnote A.

Relatedly, how does the macro-level stuff in section 5.2 help? What can it be used for? I’m not very big on macro, but I guess like GDP it might be some measure of how well-off nations are? All this is to say that if the authors spend so much time discussing stuff on large scales, they could help the reader by also explaining why this type of assessment is useful. But regardless, it seems they don’t discuss small scale valuation, which certainly has been the application of most CME valuation studies, enough.

The project-level approach is important but not the focus of this paper.  We have added the further clarification at line 622:

This alternative also does not negate the need for, and importance of, smaller scale valuation studies that link ecological production functions to specific economic benefits for coastal and marine policy decision-making [e.g. 81].

Can the authors make a more clear recommendation about the relationship between valuation and policy?  This is something I always struggle with myself, and I think the authors say it well: from an economic point of view we want to compare benefits and costs of a policy or project to see if it’s a good idea, but policy-makers don’t think the same way generally, and so any estimates economists come up with for benefits or costs (as related to CME service changes) don’t really end up playing a role in policy decisions.

This is a very broad issue and more than we can tackle in one paper.  We have added some further clarification with the statement at line 608:

Economic analysis may be difficult to apply for all CME services, but it can provide a consistent framework to develop valuation information and to evaluate that information across national boundaries.

So are valuation economists wasting their time? It gets a bit depressing, I think. I really do like how the authors highlight instances where valuation did play more of a role in policy, and what it was about the considered studies that helped them play a more effective role in policy. The authors devote a few sentences to this, but I think it’s important enough to add a few more details, like specifics of what made the studies accessible to policy-makers (if the authors want to go this route, that is). That way the reader gets some clear recommendations about how his or her work can be more relevant.

We are proposing a new direction for research that moves beyond small-scale studies.  Our intent is not to advise on existing policy processes because there are so many different situations.

The last two paragraphs before section 5.2 have a bit of circular reasoning I wish the authors could help me work through. On the one hand, using TEV “cannot account for the full range of sociocultural and ecological values from CME services” (so, it’s implied, we can’t rely solely on economic concepts of what makes a policy good or bad and we need to consider other criteria), but on the other hand, using these other non-economic criteria “[does] little to further a comprehensive understanding of values for a full range of CME services.” I guess the authors are just telling it like it is, but I feel like I want them to give me a recommendation (I being an economist) about how to deal with the reality of decision-making while still thinking like an economist and comparing costs and benefits (which requires TEV estimation).

This is also a broad issue and we are mainly trying to counter the wave of noneconomic approaches that have emerged in the ecosystem services literature.  For further clarification we added the sentence referenced earlier at line 608:

Economic analysis may be difficult to apply for all CME services, but it can provide a consistent framework to develop valuation information and to evaluate that information across national boundaries.

Other comments:

·         Table 1 is great.

Thank you.

·         p.8 line 344, It’s pretty obvious that the S in CMES stands for services, but I don’t think CMES was ever defined.

The manuscript has been revised to make sure CME services is always used.

·         p.11 line 488, I believe “relatively” should be “relative”.

Fixed.

Reviewer 3 Report

Good overview of the literature and existing work to measure and value coastal and marine ecosystem services. General comment is that this paper should do a bit better in presenting to the multidisciplinary audience terms and concepts that economists use but other disciplines don’t. In that context, the paper should also make the distinction between proper (or improper) use of valuation methods and the correct use of these methods instead.  Minor comments and additions in the text below:

Lines 38: The argument about the lack of degradation of seagrass beds is not correct. https://doi.org/10.1038/ngeo1477 show that a decline in the 20th century has been observed globally and https://doi.org/10.1016/j.scitotenv.2018.09.296 show that valuation of the ES from such ecosystems is lacking and needed.

Lines 101-103: What is missing is the service of a healthy climate from the storage of carbon in sediments and roots of species in CMEs.

Lines 120-131: This paragraph neglects the reasons behind some CMEs being more researched than others. E.g. cultural ES have been found to be more difficult to value economically due to multiple definitions of them existing https://doi.org/10.1016/j.ecoser.2017.04.002 and that benefits they provide are intangible http://dx.doi.org/10.5751/ES-05790-180344 .

Line 132: be consistent with writing seagrass or sea-grass across the document.

Lines 146: Other types of mapping exist to map marine ecosystems. Also, remote sensing can be applied in coastal ecosystems if light penetration is favourable and might even be better than in-field measuring. See https://doi.org/10.1080/01431161003692057 and https://doi.org/10.1016/j.scitotenv.2018.09.296

Lines 150-164: This paragraph has omitted an important aspect of economic valuation: values are anthropocentric. See http://dx.doi.org/10.1016/j.ecolecon.2013.09.002  For a further discussion on this point, which fuels disagreements between economists and conservationists. Also, this paragraph and the paper have mostly ignored carbon sequestration benefits (apart from lines 350-351), which is contrary to recent publication trends (e.g. https://doi.org/10.1016/j.ecoser.2018.10.013 )

To support the argument in line 283-293: see https://doi.org/10.1016/j.marpolbul.2013.03.029 for a discussion on how differences in carbon sequestration rats in seagrasses can change the benefits (calculated with the social cost of carbon or with the abatement cost method) for seagrass beds in Europe.

Lines 349-353: There have been market methods and not IAMs applied to measure the impact of climate change (.e.g. ocean acidification, see https://doi.org/10.1016/j.envsci.2018.05.008 measuring losses of landings in crustaceans and molluscs in the UK)

Lines 354-359: Although the authors have said before they do not wish to expand on the methods for valuation I suggest that at least some reference to market and non-market valuation and small description is used to help the interdisciplinary audience of the journal (preferably somewhere in section 3.1).

line 370-377: Perhaps the use of the methods and not the methods themselves is the problem the authors reference. The correct implementation and use of meta-analyses and benefit transfers has been proven beneficial in the past for the valuation of ecosystem services. See for example https://doi.org/10.1016/j.jenvman.2016.08.012 for an application for freshwater quality across Europe and https://doi.org/10.1016/j.ecolecon.2017.01.005 for a benefit transfer for 9 Baltic states. 

Lines 414-420: Consider also using https://doi.org/10.1016/j.marpol.2019.02.033 as a more recent reference for the issues mentioned here, referencing MSFD in particular.

Author Response

Good overview of the literature and existing work to measure and value coastal and marine ecosystem services. General comment is that this paper should do a bit better in presenting to the multidisciplinary audience terms and concepts that economists use but other disciplines don’t. In that context, the paper should also make the distinction between proper (or improper) use of valuation methods and the correct use of these methods instead.  Minor comments and additions in the text below:

We have added a paragraph on valuation methods based on people’s preferences vs other valuation methods to clarify this and other points:

“In general, the objective of economic valuation is to provide estimates of the net present value of ecosystem services, and a number of prior papers review the range of economic valuation methods for CMEs [e.g., 31-34].  Values for ecosystem services can be derived directly using information from prices of related market goods and services, such as by using current prices of fish products to estimate future losses in fisheries due to ocean acidification [e.g., 35].  However, most ecosystem services benefit humans in ways that are not captured in existing markets, so non-market valuation methods are necessary for these cases.  Broadly, non-market valuation methods can be divided into revealed preference and stated preference methods.  With revealed preference methods such as the travel cost and hedonic pricing methods, information on human behavior in existing markets, such as travel or housing, are used to estimate the value of related ecosystem services, such as beach recreation (by observing preferences for people to travel to beaches) or clean waterways (by observing price premiums for homes adjacent to clean water).  In contrast, stated preference methods such as contingent valuation or discrete choice experiments rely on surveys designed to elicit people’s preferences for specific ecosystem services.  In recent years, studies have also used estimates of the ‘social cost of carbon’ to build estimates of the net present value of services related to carbon storage.  The major difference between this approach and the market and non-market methods is that the social cost of carbon approach does not rely on studying people’s preferences, but rather on simulations of the global economy under different climate change scenarios.

Lines 38: The argument about the lack of degradation of seagrass beds is not correct. https://doi.org/10.1038/ngeo1477 show that a decline in the 20th century has been observed globally and https://doi.org/10.1016/j.scitotenv.2018.09.296 show that valuation of the ES from such ecosystems is lacking and needed.

We have incorporated the decline of seagrass beds in the introduction.

Lines 101-103: What is missing is the service of a healthy climate from the storage of carbon in sediments and roots of species in CMEs.

We have added carbon storage to the list of regulating services provided by CMEs.

Lines 120-131: This paragraph neglects the reasons behind some CMEs being more researched than others. E.g. cultural ES have been found to be more difficult to value economically due to multiple definitions of them existing https://doi.org/10.1016/j.ecoser.2017.04.002 and that benefits they provide are intangible http://dx.doi.org/10.5751/ES-05790-180344.

These studies and the difficulties inherent in valuing cultural ecosystem services are now referenced in section 3, where similar difficulties with the valuation of biodiversity are also discussed.

Line 132: be consistent with writing seagrass or sea-grass across the document.

Thank you.  Corrected and kept seagrass.

Lines 146: Other types of mapping exist to map marine ecosystems. Also, remote sensing can be applied in coastal ecosystems if light penetration is favourable and might even be better than in-field measuring. See https://doi.org/10.1080/01431161003692057 and https://doi.org/10.1016/j.scitotenv.2018.09.296

Thank you, we have added a sentence with references pointing to these developments in remote sensing methods.

Lines 150-164: This paragraph has omitted an important aspect of economic valuation: values are anthropocentric. See http://dx.doi.org/10.1016/j.ecolecon.2013.09.002  For a further discussion on this point, which fuels disagreements between economists and conservationists.

We have added the following sentence referencing this issue:

In other words, TEV is an anthropocentric concept that considers economic value strictly as physical or perceived benefits to humans, a point which has fueled disagreements between economists and conservationists on the issue of intrinsic value [25].

Also, this paragraph and the paper have mostly ignored carbon sequestration benefits (apart from lines 350-351), which is contrary to recent publication trends (e.g. https://doi.org/10.1016/j.ecoser.2018.10.013 )

We have added multiple references and mentions of the carbon storage benefits of CMEs.  However, note that valuing storage benefits by using the social cost of carbon is different from valuing ecosystem services by estimating demand curves for ecosystem services.  Our paper is a review of the latter.

We have added a paragraph on valuation methods based on people’s preferences vs other valuation methods to clarify this and other points.

To support the argument in line 283-293: see https://doi.org/10.1016/j.marpolbul.2013.03.029 for a discussion on how differences in carbon sequestration rates in seagrasses can change the benefits (calculated with the social cost of carbon or with the abatement cost method) for seagrass beds in Europe.

We have added a reference to this paper when discussing studies valuing carbon storage benefits using the social costs of carbon.

Lines 349-353: There have been market methods and not IAMs applied to measure the impact of climate change (.e.g. ocean acidification, see https://doi.org/10.1016/j.envsci.2018.05.008 measuring losses of landings in crustaceans and molluscs in the UK)

We have added a reference to this paper as one of the exceptions—a study that does not use IAMs or the social cost of carbon to value the impacts of climate change.

Lines 354-359: Although the authors have said before they do not wish to expand on the methods for valuation I suggest that at least some reference to market and non-market valuation and small description is used to help the interdisciplinary audience of the journal (preferably somewhere in section 3.1).

We have added a paragraph on valuation methods based on people’s preferences vs other valuation methods to clarify this and other points.

line 370-377: Perhaps the use of the methods and not the methods themselves is the problem the authors reference. The correct implementation and use of meta-analyses and benefit transfers has been proven beneficial in the past for the valuation of ecosystem services. See for example https://doi.org/10.1016/j.jenvman.2016.08.012 for an application for freshwater quality across Europe and https://doi.org/10.1016/j.ecolecon.2017.01.005 for a benefit transfer for 9 Baltic states.

Our paper is not criticizing benefit transfer as a way to extrapolate values for specific ecosystem goods or services, as was done in the above referenced studies.  These studies, for example, use BT to extrapolate well-understood values for water quality (a very specific good) from one population to another.  In contrast, studies such as Costanza et al. intend to use this methodology to extrapolate values for ALL of the world ecosystems, and values for many of these ecosystems are unknown or poorly understood.

Lines 414-420: Consider also using https://doi.org/10.1016/j.marpol.2019.02.033 as a more recent reference for the issues mentioned here, referencing MSFD in particular.

Thank you, this is a great reference. We have added as reference 63 and are using it in multiple parts of the manuscript.